# Task-Aware Self-Supervised Framework for Dialogue Discourse Parsing

**Wei Li[a], Luyao Zhu[a], Wei Shao[b], Zonglin Yang[a]** and **Erik Cambria[a]**

[a]Nanyang Technological University, Singapore
[b]City University of Hong Kong, Hong Kong SAR
{wei008, luyao001, zonglin001}@e.ntu.edu.sg
weishao4-c@my.cityu.edu.hk, cambria@ntu.edu.sg

## Abstract

Dialogue discourse parsing is a fundamental natural language processing task. It can benefit a series of conversation-related downstream tasks including dialogue summarization and emotion recognition in conversations. However, existing parsing approaches are constrained by predefined relation types, which can impede the adaptability of the parser for downstream tasks. To this end, we propose to introduce a task-aware paradigm to improve the versatility of the parser in this paper. Moreover, to alleviate error propagation and learning bias, we design a graph-based discourse parsing model termed DialogDP. Building upon the symmetrical property of matrix-embedded parsing graphs, we have developed an innovative self-supervised mechanism that leverages both bottom-up and top-down parsing strategies. This approach allows the parsing graphs to mutually regularize and enhance each other. Empirical studies on dialogue discourse parsing datasets and a downstream task demonstrate the effectiveness and flexibility of our framework[1].

## 1 Introduction

Dialogue discourse parsing (DDP) plays an essential role in the field of natural language processing (NLP), serving as a foundational task and receiving increasing attention from the research community (Shi and Huang, 2019; Yang et al., 2021; Yu et al., 2022). The formulation of the DDP task is rooted in the Segmented Discourse Relation Theory (Asher and Lascarides, 2003), distinguishing it from the Rhetorical Structure Theory (Mann and Thompson, 1988) and the Penn Discourse Relation Theory (Prasad et al., 2008) that primarily underpins text-level discourse parsing (Afantenos et al., 2015). The primary objective of DDP is to recognize the links and relations between utterances in dialogues.

[1] https://github.com/senticnet/DialogDP

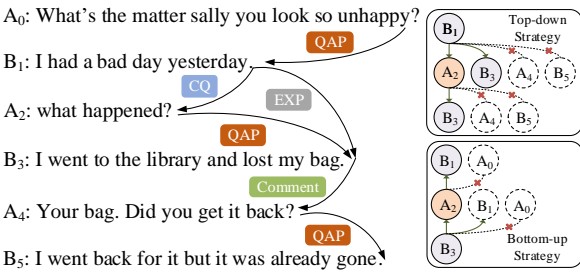

Figure 1: An example of a dialogue dependency graph for a dailydialog (Li et al., 2017) conversation snippet. Here *A* and *B* are two speakers with subscripts indicating the turn of the dialogue. "QAP", "CQ", "EXP" and "Comment" stand for dependency relations.

The parsed dependencies between utterances should form a Directed Acyclic Graph (DAG). For example, dependent utterance $B_3$ in Fig. 1 depends on two head utterances $B_1$ and $A_2$ in "Explanation" and "QAP" relations, respectively. The early neural-based paradigm (Shi and Huang, 2019) for the DDP task involved sequentially scanning the utterances in dialogues and subsequently predicting dependency links and corresponding relation types, which is prone to the severe error propagation issue (Wang et al., 2021), as the learned representations relied on historical predictions. This observation inspired us to formalize DDP from a graph perspective, where the prediction of links and relations for each pair of utterances is independent of the others.

Existing methods (Wang et al., 2021) resorted to the bottom-up strategy for DDP, where for each dependent utterance, the parser retrieved only one head utterance. These models (He et al., 2021) may suffer from measurement bias (Mehrabi et al., 2021) as the training labels are distorted with the one-head retrieval for samples with multi-head labels. Furthermore, these multi-head labels serve as long-tailed data, and overlooking the long-tailed data may result in a biased distribution estimation during training (Wang et al., 2022).

It inspires us to keep the multi-head labels for bottom-up strategy training. Moreover, Yang et al. (2021) and Yu et al. (2022) used external knowledge or other NLP tasks to mitigate algorithmic bias (Mehrabi et al., 2021) during training. Nevertheless, they omitted the potential of leveraging internal structure for regularization. While the top-down strategy can be effective in sentence parsing (Koto et al., 2021), it was rarely exploited in DDP due to the requirement of predicting multiple dependents for each head in dialogues. Despite the opposite directions of the two strategies, there exists a strong correlation between the relevance scores of the same utterance pair in both strategies. It leaves room for reciprocal structural supervision between bottom-up and top-down strategies.

Meanwhile, as a foundational NLP task, DDP has proved beneficial for downstream tasks including dialogue summarization (Chen and Yang, 2021) and emotion recognition in conversations (ERC) (Li et al., 2022, 2023). Nevertheless, existing parsers are constrained by predefined relation types, posing a potential obstacle to the parser's adaptability for downstream tasks. For example, existing relations like narration and background do not/ or rarely exist in the ERC datasets. In addition, a well-designed DDP fine-tuned with task-aware dependency labels can capture emotion shifts, which benefits the downstream ERC task.

Overall, the previous methods have three limitations, i.e., error propagation, learning bias of distorted training labels and a single strategy, and incompatibility of predefined relations with downstream tasks. To this end, we propose a task-aware self-supervised framework for DDP task. Concretely, a graph-based model DialogDP, utilizing biaffine mechanism (Dozat and Manning, 2017), is designed for DDP, avoiding sequential error propagation. The model consists of symmetric parsers, simultaneously performing bottom-up and top-down parsing.

We investigate parsed links and relation graphs of the two strategies and design a bidirectional self-supervision mechanism encouraging the two strategies to acquire similar graphs through mutual learning. Moreover, we propose a soft-window triangular (SWT) mask incorporating a soft constraint, as opposed to a hard constraint (Shi and Huang, 2019), to guide the parsers. SWT mask encourages parsers to prioritize candidate links within a flexible window for each utterance.

To enhance the adaptability of DialogDP for downstream tasks, we propose a novel paradigm involving fine-tuning with task-specific re-annotated relations. We validate the effectiveness of our task-aware paradigm on downstream ERC. The contributions of this paper are:

- We propose a new DDP model that explicitly captures structures of dependency graphs with bottom-up and top-down strategies, avoiding sequential error propagation.

- Bidirectional self-supervision with an SWT mask is devised to alleviate the learning bias.

- Our parser surpasses baselines on benchmark datasets and task-aware DialogDP demonstrates superior effectiveness in handling downstream tasks.

## 2 Related Work

DDP aims to analyze a conversation between two or more speakers to recognize the dependency structure of a dialogue. Compared with the general text-level discourse parsing (Mann and Thompson, 1988; Prasad et al., 2008; Li et al., 2014b; Afantenos et al., 2015), DDP provides significant improvement for many dialogue-related downstream tasks (Ma et al., 2023; Zhang and Zhao, 2021; Chen and Yang, 2021) via introducing symbolic dialogue structure information into the modeling process.

Existing works mainly focused on applying neural models to handle problems in DDP. In detail, Shi and Huang (2019) formalized DDP as a dependency-based rather than a constituency-based (Li et al., 2014a) parsing. However, their sequential scan method introduced error propagation. Wang et al. (2021) proposed a structure self-aware model, producing representations independently of historical predictions to handle error propagation, yet still encounters learning bias. Recent methods used external knowledge or other NLP tasks to mitigate bias during training. Liu and Chen (2021) utilized domain adaptation techniques to produce enhanced training data and thus improved the DDP model's performance. Yu et al. (2022) pointed out the lack of modeling speaker interactions in previous works and proposed a joint learning model. Fan et al. (2022) presented a distance-aware multitask framework combining both transition- and graph-based paradigms. Nevertheless, they overlooked the potential of leveraging the internal structure for regularization.

In summary, despite the good performance, the existing methods still have problems in modeling and application. Modeling-wise, the DDP is currently limited to either a top-down or bottom-up manner, leading to a gap in achieving bidirectionality. Application-wise, the issue arises due to the constraints imposed by predefined relation types, thereby limiting the benefits to only tasks directly associated with these pre-defined relation types. Consequently, there is a need to establish connections between the parsed dependency relations and downstream tasks. In this paper, we propose a new framework to address these important issues.

## 3 Methodology

DialogDP is tailored for integral graph-driven discourse parsing, which is more computation-efficient and avoids error propagation in the training process. Specifically, the bidirectional self-supervision allows the parser to parse the dialogue with both top-down and bottom-up strategies. The employed strategies in both directions can mutually reinforce and guide each other. Moreover, the SWT mask guides the two strategies and imposes soft constraints on the integral learned graph, prompting the parser to prioritize the candidate links of each utterance within a flexible window.

### 3.1 Task Definition

Given a multi-turn multi-party (or dyadic) dialogue $U$ consists of a sequence of utterances $\{u_1, u_2, ..., u_n\}$, the goal of the DDP task is to identify links and the corresponding dependency types $\{(u_j, u_i, r_{ji})|j \neq i\}$ between utterances, where $(u_j, u_i, r_{ji})$ represents a dependency with type $r_{ji}$ from dependent utterance $u_j$ to head utterance $u_i$. We formulate the DDP as a graph spanning process, where the dependency link of the current utterance $u_i$ is predicted by calculating a probability distribution $P(u_j|u_i, j \neq i)$ over other utterances. Dependency relation type prediction is formulated as a multi-class classification task, where the probability distribution is computed as $P(t|u_i, u_j, t \in C)$. $|C|$ is the number of pre-defined relation types.

The parsed dependencies between utterances constitute a DAG (Shi and Huang, 2019). However, due to the limited presence of multiple incoming relations in STAC and Molweni datasets, most existing methods parse a dependency tree (Liu and Chen, 2021) which is a special type of DAG. The dependency types are predefined as 16 relations (Appendix A), specified by (Asher et al., 2016). Following (Li et al., 2014b), we add a *root* node, denoted as $u_0$. An utterance is linked to $u_0$, if not connected to preceding utterances.

### 3.2 Model Overview

To tackle error propagation in sequential scan, we re-formalized the DDP in a graph-based manner, where a link graph and a dependency-type graph are built based on the scores computed by the biaffine (Dozat and Manning, 2017; Zhang et al., 2020) mechanism. The parsed dependency tree can be obtained by jointly decoding the two graphs. Fig. 2 illustrates the structure of our proposed DialogDP model. **First**, the pre-trained large language model BERT (Devlin et al., 2018) is employed to generate speaker-aware and context-aware utterance-level representations. **Second**, a bidirectional self-supervision mechanism is designed to capture the links and relations between utterances, and an SWT mask is applied to regularizing the learned graphs in an explainable manner.

### 3.3 Speaker-Context Encoder

Given a dialogue $U = \{u_1, u_2, ..., u_n\}$ with a corresponding speaker list $Sp = \{Sp_1, Sp_2, ..., Sp_n\}$, we concatenate the whole dialogue in a single sequence $x = \{[cls], [sp_1], u_1, [sp_2], u_2, ..., [sp_n], u_n\}$, where $[cls]$ and $[sp_i]$ are special tokens of start of the sequence and speaker $Sp_i$, respectively. Then speaker-context integrated embeddings $e$ can be obtained through $PLM(x)$. $e_i := PLM(x)_j$ is the speaker-context integrated embedding of utterance $u_i$ of speaker $Sp_i$, where $j$ is obtained when $x_j = [sp_i]$.

### 3.4 Bidirectional Self-Supervision

DialogDP is composed of two symmetric components, a bottom-up parser and a top-down parser. The two parsers are designed on the basis of the biaffine mechanism (Dozat and Manning, 2017) which is proved to be effective on sentence-level dependency parsing. The bottom-up strategy involves the parser calculating the biaffine attention score between a dependent and a head, and selecting the head with the highest score for each dependent. In contrast, the top-down parser identifies the dependents with high scores for each head. Subsequently, both parsers build a link graph and a relation graph, wherein each node represents an utterance, and the arcs connect pairs of nodes within the graph.

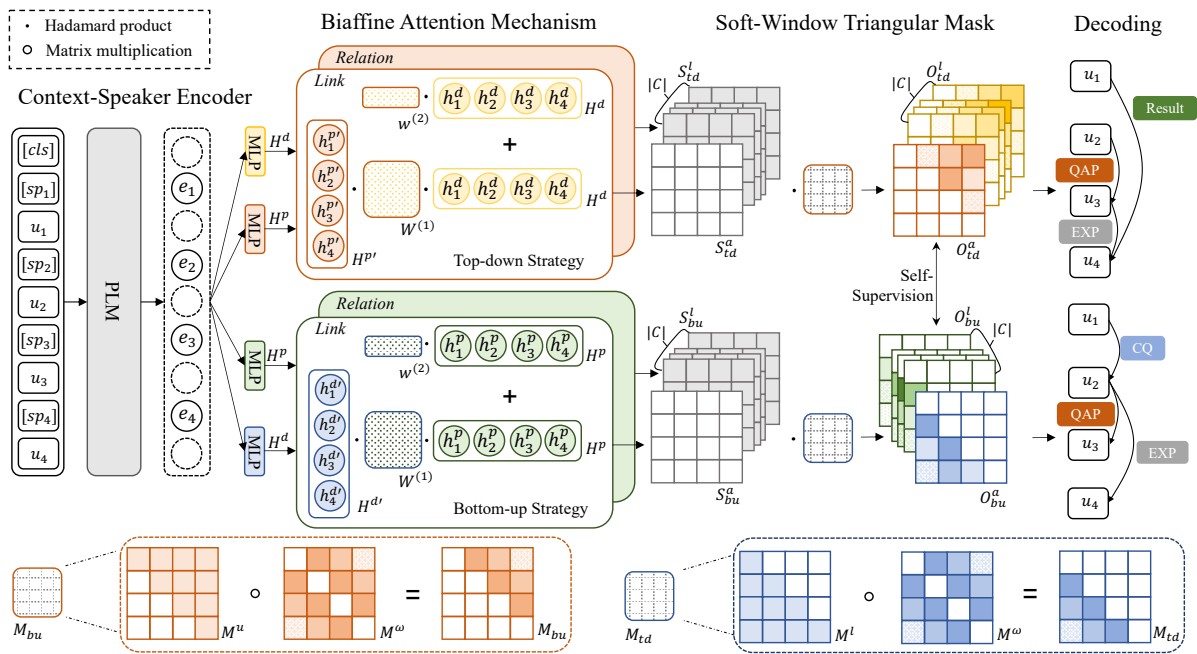

Figure 2: Architecture of DialogDP.

We calculate the arc scores ($S_{bu}^a = [s_1^a, s_i^a, ..., s_n^a]_{bu}$) and relation label scores ($S_{bu}^l = [s_1^l, s_i^l, ..., s_n^l]_{bu}$) with bottom-up strategy through:

$$s_i^a = h_i^{d'} W_a^{(1)} H^p + w_a^{(2)} H^p \quad (1)$$

$$s_{i,j}^l = r_i^{d'} \mathbf{W}_l^{(1)} r_j^p + (r_j^p \oplus r_i^d)' W_l^{(2)} + b. \quad (2)$$

Here, $H^p \in \mathbb{R}^{d \times n}$ and $r_{y_i}^p \in \mathbb{R}^{d \times 1}$ are the head hidden states from two different multiple layer perceptrons (MLPs), while $h_i^{d'} \in \mathbb{R}^{1 \times d}$ and $r_i^{d'} \in \mathbb{R}^{1 \times d}$ are the dependent hidden states from another two different MLPs. $s_i^a \in \mathbb{R}^{1 \times n}$ is the attention scores between the $i$-th dependent and $n$ heads. $s_i^l \in \mathbb{R}^{1 \times |C|}$ is the likelihoods of each class given the $i$-th dependent and $n$ heads. Similarly, the arc scores ($S_{td}^a$) and relation label scores ($S_{td}^l$) with top-down strategy can be obtained by exchanging the positions of head hidden states and dependent hidden states in the above formulas. $w_a^{(2)} \in \mathbb{R}^{1 \times d}$, $W_a^{(1)}, W_l^{(2)} \in \mathbb{R}^{d \times d}$ and $\mathbf{W}_l^{(1)} \in \mathbb{R}^{d \times |C| \times d}$ are learnable parameters of the biaffine mechanism.

By observing the arc (relation) scores of both the top-down strategy and the bottom-up strategy, we can readily infer their high relevance. In other words, $S_{bu}^a \propto S_{td}^{a\,'}$ and $S_{bu}^l \propto S_{td}^{l\,'}$. Hence, we designed a bidirectional self-supervision mechanism for regularizing the bidirectional strategies by assuming $S_{bu}^a = S_{td}^{a\,'}$ and $S_{bu}^l = S_{td}^{l\,'}$.

The symmetric self-supervision losses are

$$\mathcal{L}_s^a = KL(S_{bu}^a || S_{td}^{a\,'}) + KL(S_{td}^{a\,'} || S_{bu}^a) \quad (3)$$

$$\mathcal{L}_s^l = KL(S_{bu}^l || S_{td}^{l\,'}) + KL(S_{td}^{l\,'} || S_{bu}^l). \quad (4)$$

Here $KL(P||Q) = \sum_{n \times n} P \log(\frac{P}{Q})$ is the Kullback–Leibler divergence (Kullback and Leibler, 1951) between the two distributions. $P$ and $Q$ refer to the aforementioned paired attention scores.

### 3.5 Soft-Window Triangular Mask

In this work, we assume that all the head utterances appear before the dependent utterances. This aligns with our intuition that the current utterance cannot be induced by the preceding utterance. As we can see in Fig. 2, the feasible attention scores, for bottom-up strategy, between each head and the candidate dependents distributed in the lower triangular matrix $M^l = [m_{ij}^l]$ such that $m_{ij}^l = 0$ for $i \leq j$, as bottom-up strategy is to retrieve one head for each dependent. Similarly, top-down strategy corresponds to the upper triangular matrix $M^u = [m_{ij}^u]$ such that $m_{ij}^u = 0$ for $i \geq j$, as it identifies the dependents for each head.

The previous method (Shi and Huang, 2019) implemented a hard window constraint, compelling the model to exclusively select links within a predetermined distance range. However, the hard window excluded all candidates whose distance from the current utterance exceeds the predefined window size.

This limitation hampers the generalization capability in real-world scenarios.To this end, we devise a soft-window mechanism that can be implemented by a carefully designed mask. Each element $m_{i,j}^{\omega}$ of the mask $M^{\omega}$ is calculated by:

$$m_{i,j}^{\omega} = \begin{cases} 1, & d_{i,j} < B_0 \\ f(\log(P_{i,j}), \alpha_0, 1), & B_0 \leq d_{i,j} < B_1 \\ f(\log(P_{i,j}), \alpha_1, \alpha_0), & B_1 \leq d_{i,j} < B_2 \\ \alpha_2, & d_{i,j} \geq B_2 \end{cases}$$

where $f(\cdot, min, max)$ is the Max-Min scaling that scales each input feature to a given range $[min, max]$. $d_{i,j}$ denotes $|i - j|$. $B_0$, $B_1$, and $B_2$ are the distance boundaries suggested by the prior distribution of head-dependent distances, of which the distribution plot is in Appendix A. $\alpha_0 = 0.95$, $\alpha_1 = 0.85$ and $\alpha_2 = 0.7$ are the hyper-parameters.

Combining the triangular mask with the soft-window mask, SWT masks for bottom-up and top-down strategies, $M_{bu}$ and $M_{td}$ are obtained by

$$M_{bu} = M^l M^{\omega} \quad (5)$$
$$M_{td} = M^u M^{\omega}. \quad (6)$$

The scores of arcs and relations after the soft-window triangular masking are $O^a \in \mathbb{R}^{n \times n}$ and $O^l \in \mathbb{R}^{n \times n \times |C|}$, respectively. We denote $o_{i,j}^a := [O^a]_{i,j}$ and $o_{i,j,k}^l := [O^l]_{i,j,k}$.

### 3.6 Link and Relation Decoding

In the decoding process, the parser leverages the bottom-up strategy to predict the link and the corresponding relations. Concretely, for each dependent utterance $u_i$, the parser predicts its head $u_j$ and then recognizes the relation of the predicted link.

$$P(\hat{y}_{i,j} = 1 | u_i, U_{<i}) = \frac{\exp(o_{i,j}^a)}{\sum_{k<i} \exp(o_{i,k}^a)} \quad (7)$$

$\hat{y}_{i,j}$ denotes whether the parser predicts a link between $u_i$ and $u_j$. $k < i$ indicates that the predicted head should be the preceding utterance before the current dependent utterance, which is achieved by applying the triangular mask. Then, the $j$-th utterance is determined as the head through

$$j = \arg\max_{j<i} P(\hat{y}_{i,j} = 1 | U_i, U_{<i}). \quad (8)$$

Similarly, the parser predicts the relation between $u_i$ and its selected head $u_j$. The probability distribution over all relations is calculated as follows:

$$P(t | u_i, u_j, t \in C) = \frac{\exp(o_{i,j,t}^l)}{\sum_{c \in C} \exp(o_{i,j,c}^l)}. \quad (9)$$

### 3.7 Loss Function

We use binary cross entropy (BCE) loss for multi-label classification as link prediction loss for both strategies, and adopt cross entropy (CE) loss for relation classification. Here we show the classification loss functions of the top-down strategy, as the classification losses of both strategies are similar.

$$\mathcal{L}_{td}^a = -\frac{1}{n(n-i)} \sum_i^n \sum_{j>i}^n \text{BCE}(o_{i,j}^a, y_{i,j}) \quad (10)$$

$$\mathcal{L}_{td}^l = -\frac{1}{n_l |C|} \sum_{y_{i,j}=1} \sum_{c \in C} \text{CE}(o_{i,j}^l, y_{i,j}^l) \quad (11)$$

For $u_i$ and $u_j$, $y_{i,j}$ is a binary label for link and $y_{i,j}^l$ is a relation label. Here $n_l$ is the number of existing links in a dialogue. While the bottom-up strategy in decoding selects just one head for each dependent, it is trained using a multi-label classification approach. This is essential because a small set of dependents may have multiple ground-truth heads. Our method capitalizes on this to fully exploit the instructive dependent-head label information, which is often ignored by other methods. In summary, the total loss of the framework is the weighted summation of classification and supervision losses.

$$\mathcal{L} = \mathcal{L}_{bu}^a + \mathcal{L}_{bu}^l + \mathcal{L}_{td}^a + \mathcal{L}_{td}^l + \lambda_a \mathcal{L}_s^a + \lambda_l \mathcal{L}_s^l \quad (12)$$

Here $\lambda_a$ and $\lambda_l$ are trade-off parameters.

### 3.8 Task-Aware Dialogue Discourse Parsing

Previous researches on DDP followed the original definition of 16 relations, regardless of application scenarios. However, the existing relation taxonomy may not fit the downstream tasks, e.g., ERC. Hence, we propose a task-aware DDP paradigm to improve its adaptability to downstream tasks. In Fig. 3, the task-aware DialogDP bridges the gap between the fundamental DDP with downstream tasks through fine-tuning DialogDP with task-aware relation annotations. The main steps include:

(a) Train DialogDP on existing DDP datasets with the default taxonomy of dependency relations.

(b) Predict dependency links $A$ of the dialogue $U^d$ in the downstream task training dataset.

(c) Automatically re-annotate relations $T^d$ for predicted links $A$ with rule-based methods based on target labels for the downstream task, which is detailed in Algorithm 1.

(d) Fine-tune DialogDP on downstream training dataset with $A$ and $T^d$.

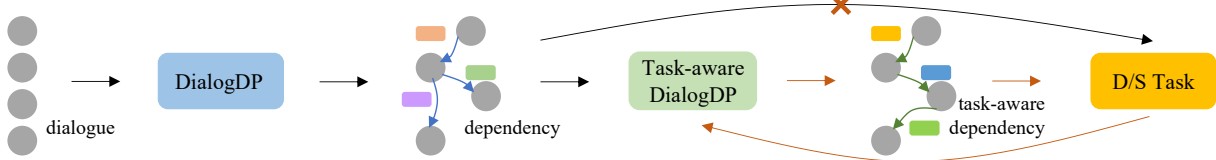

Figure 3: Framework of task-aware DDP. D/S denotes downstream. The ✗ means not using the predefined relation labels. The backward arrow from D/S Task to Task-aware DialogDP corresponds to step (c) in subsection 3.8.

(e) Predict dependency links and relations with DialogDP for test dataset and incorporate results into input features of downstream tasks.

---

**Algorithm 1:** Task-aware annotation

**Data:** downstream dialogue $U^d$ with labels $L \in \mathbb{N}^+ \cup \{0\}$, predicted links $Y$
**Result:** task-aware relations $T^d$
1 define state transition mapping $g(l_i, l_j), (l_i, l_j \in L)$;
2 **while** $y_{i,j}$ *in* $Y$ *and* $y_{i,j} \neq 0$ **do**
3 $\quad$ $t_{i,j}^d \leftarrow g(l_i, l_j)$;
4 **end**

---

Taking the downstream ERC task as an example, task-aware DDP leverages the state transition to represent emotion shift which can enhance ERC performance (Gao et al., 2022) and contribute to the research in the psychology domain (Winkler et al., 2023). The state transition mapping $g$ (eq. 13) from head $u_i^d$ to $u_j^d$ is obtained based on their target labels $l_i$ and $l_j$, where $l_i, l_j \in \{0, 1, 2\}$ corresponding to {*negative*, *neutral*, *positive*} sentiment polarities.

$$g(l_i, l_j) = \begin{pmatrix} 0 & 1 & 2 \\ 3 & 0 & 4 \\ 5 & 1 & 0 \end{pmatrix} \quad (13)$$

The numbers from 0 to 5 in $g$ are the relation types for ERC task-aware DDP, referring to "same_polarity", "sentiment_to_neutral", "negative_to_positive", "neutral_to_negative", "neutral_to_positive", and "positive_to_negative", respectively. In this case, the task-aware DialogDP fine-tuned with re-annotated relations is capable of capturing sentiment shifting between utterances.

## 4 Experiment Setups

### 4.1 Dataset

We evaluated DialogDP on two publicly available datasets, i.e., STAC[2] (Asher et al., 2016) and Molweni[3] (Li et al., 2020). **STAC** is collected from

an online game, *The Settlers of Catan*. It consists of $1,173$ annotated dialogues, divided into two sets, i.e., $1,062$ dialogues for training and 111 for testing. **Molweni** is collected from Ubuntu Chat Corpus (Lowe et al., 2015). The dataset consists of $9,000$ annotated training instances, along with 500 instances allocated for development and another 500 instances for testing. We pre-processed datasets following (Shi and Huang, 2019).

### 4.2 Setups and Metrics

We used BERT-large from HuggingFace[4] as the speaker-context encoder. The optimizer is AdamW (Loshchilov and Hutter, 2018) with initial learning rates of 1e-5 and 5e-6 for DialogDP and BERT encoder, respectively. The maximum value of gradient clipping is 10. Both $\lambda_a$ and $\lambda_l$ are configured with 0.05. The dropout rate is configured to 0.33, adhering to the default setting of biaffine. Following Shi and Huang (2019), we report micro F1 scores for LINK prediction and L&R prediction, respectively. In L&R, both the link and its corresponding relation should be correctly classified. The experiments were performed on a V100 GPU with 16 GB of memory. To provide accurate results, we conducted three random runs on test sets and reported the average score.

### 4.3 Baselines

Shi and Huang (2019) proposed DeepSequential, which sequentially scans the utterance and predicts the link and its corresponding relation type sequentially. Wang et al. (2021) presented a structure self-aware model, which adopts an edge-centric graph neural network. Liu and Chen (2021) put forward a framework leveraging cross domain data to improve the generalization ability of the neural parser. Li et al. (2020) built the Molweni dataset and proposed a parser on the basis of DeepSequential. He et al. (2021) advanced a multitask learning framework to jointly learn question-answering and discourse parsing tasks.

---

[2]https://www.irit.fr/STAC/corpus.html
[3]https://github.com/HIT-SCIR/Molweni

[4]https://github.com/huggingface/transformers

| MODELS | STAC | | Molweni | |
| --- | --- | --- | --- | --- |
| | Link | L&R | Link | L&R |
| Shi and Huang (2019) | 73.2 | 55.7 | 77.9 | 54.4 |
| Li et al. (2020) | – | – | 78.1 | 54.8 |
| Wang et al. (2021) | 73.5 | 57.3 | 81.6 | 58.5 |
| Liu and Chen (2021) | **75.3** | 56.9 | 79.7 | 55.9 |
| He et al. (2021)* | – | – | 80.0 | 57.0 |
| Yang et al. (2021) | 74.1 | 57.0 | – | – |
| Fan et al. (2022) | 73.6 | 57.4 | 82.5 | 58.9 |
| Yu et al. (2022) | 73.0 | 57.4 | **83.7** | 59.4 |
| DialogDP | 73.0 | **58.5** | 83.2 | **59.8** |

Table 1: Main results on STAC and Molweni datasets. "*" denotes the re-run results. L&R refers to LINK&RELATION. The bold text reveals the best performance and the underlined indicates the second best.

| MODELS | STAC | | Molweni | |
| --- | --- | --- | --- | --- |
| | Link | L&R | Link | L&R |
| DialogDP | 73.0 | 58.5 | 83.2 | 59.8 |
| w/o Link_Sup | $72.0_{\downarrow1.0}$ | $57.0_{\downarrow1.5}$ | $83.1_{\downarrow0.1}$ | $58.5_{\downarrow1.3}$ |
| w/o Rel_Sup | $72.5_{\downarrow0.5}$ | $55.6_{\downarrow2.9}$ | $82.1_{\downarrow1.1}$ | $58.7_{\downarrow1.1}$ |
| w/o L&R_Sup | $72.0_{\downarrow1.0}$ | $56.1_{\downarrow2.4}$ | $83.5_{\uparrow0.3}$ | $58.5_{\downarrow1.3}$ |
| w/o SWM | $72.4_{\downarrow0.6}$ | $55.8_{\downarrow2.7}$ | $83.0_{\downarrow0.2}$ | $58.7_{\downarrow1.1}$ |
| Biaffine | $71.2_{\downarrow1.8}$ | $55.3_{\downarrow3.2}$ | $82.5_{\downarrow0.7}$ | $58.1_{\downarrow1.7}$ |
| $Sup_{BU}$ | $71.5_{\downarrow1.5}$ | $55.5_{\downarrow3.0}$ | $82.9_{\downarrow0.3}$ | $58.5_{\downarrow1.3}$ |
| $Sup_{TD}$ | $71.9_{\downarrow1.1}$ | $56.5_{\downarrow2.0}$ | $83.1_{\downarrow0.1}$ | $58.1_{\downarrow1.7}$ |

Table 2: Ablation study results. Link_Sup, Rel_Sup, L&R_Sup and SWM corresponds to link supervision, relation supervision, link and relation supervision, and soft-window mask, respectively. Biaffine refers to a variant without self-supervision or soft-window mask. $Sup_{BU}$ ($Sup_{TD}$) refers to a single supervision mechanism where only bottom-up (top-down) strategy is supervised by top-down (bottom-up) strategy.

Yang et al. (2021) presented a unified framework DiscProReco to jointly learn dropped pronoun recovery and DDP. Yu et al. (2022) proposed a speaker-context interaction joint encoding model, taking the interactions between different speakers into account. Fan et al. (2022) combined the advantages of both transition- and graph-based paradigms.

### 4.4 Main Results

We report the DDP results of our DialogDP and baselines on STAC and Molweni datasets. In Table 1, DialogDP outperforms all the baselines on L&R and achieves comparable LINK prediction results. We believe that the weak L&R of baselines can be attributed to two factors, i.e., link predictor and relation classifier. The F1 scores on Molweni demonstrate that the majority of the baselines exhibit weaknesses in both link prediction and relation classification. The results on STAC indicate that some baselines have a relatively stronger link predictor yet obtain a poor performance on L&R. This is because their weaker relation classifiers may fail to identify the relations of predicted links, even if the links are determined accurately. Shi and Huang (2019) set a fixed window in their model, which reduced the complexity of link prediction in DDP yet compromised the model's capacity to capture relations in long-range dependencies.

### 4.5 Ablation Study

We conducted ablation studies to further explore the functions of essential components in DialogDP. Table 2 demonstrates that the excellent performance of DialogDP can be attributed to the inclusion of the proposed bidirectional self-supervision

mechanism and soft-window mask. Specifically, if L&R_Sup is removed from DialogDP, the F1 scores on the four indices drop 1.1% on average. Without SWM, the performance decreases 1.15% on average. The results of the Biaffine model highlight the significance of incorporating both bidirectional self-supervision and SWM mechanisms. The average F1 score drops of w/o Link_Sup on both LINK(0.55%) and L&R(1.4%) reveals that link supervision benefits both link prediction and overall dependency parsing. This observation is in line with our intuition, as the accuracy of subsequent relation classification depends on that of the predicted links. Furthermore, we observe that the inclusion of relation supervision not only improves link prediction but also suggests a reciprocal influence between link prediction and relation classification. The results of $Sup_{BU}$ and $Sup_{TD}$ prove DialogDP is regularized by the bidirectional self-supervision, which reduces the training bias.

### 4.6 TA-DialogDP on ERC task

We conducted experiments on an ERC dataset MELD (Poria et al., 2019) to investigate the proposed task-aware paradigm. Specifically, we selected SKIER (Li et al., 2023) as the backbone to verify the effectiveness of the generated dialogue dependency graph by the task-aware mechanism, as it explicitly leveraged parsed trees for ERC. In Table 3, the DialogDP-based models significantly outperform SKIER $_{w/o TL}$ and DialogDP models in task-aware setup even achieve comparable results with SKIER, especially on 3-class sentiment analysis.

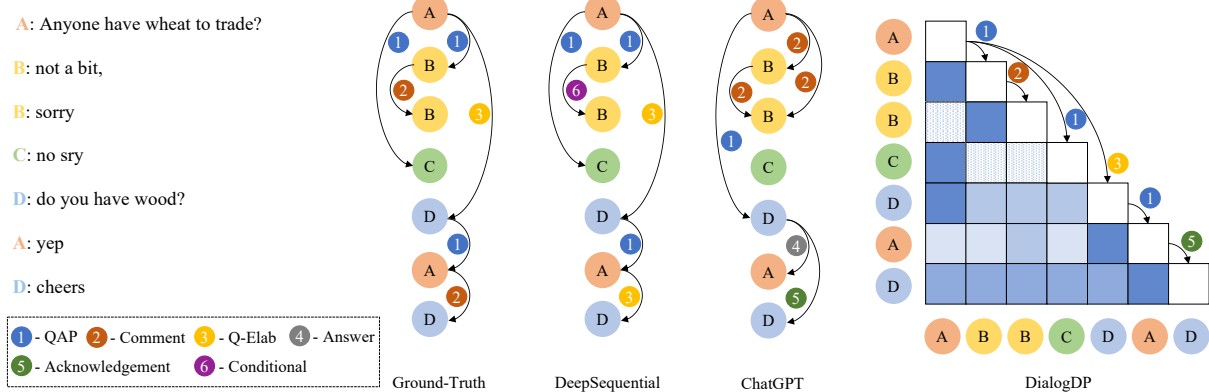

Figure 4: Case studies of on a STAC dialogue. In the lower triangular attention matrix of DialogDP, the shades varying from light blue to dark blue depict the link prediction scores ranging from 0 to 1.

Our observations indicate, in general, the task-aware setup provides a slight benefit to DialogDP, except for TA-DialogDP$_{(STAC)}$ in 7-class ERC task. We believe this may be attributed to the fact that the task-aware relation annotation specifically focuses on polarity shifting without differentiating emotion shifting. We also employed ChatGPT C for ERC. Overall, the performance of ChatGPT lags significantly behind that of SKIER, primarily due to the zero-shot setting. However, the inclusion of task-aware prompts helps to align the ChatGPT with downstream ERC through task-specific relations, whereas prompts integrated with DDP may present a challenge for ChatGPT in comprehending fundamental relation types.

| Methods | MELD | |
|---|---|---|
| | 3-cls | 7-cls |
| ChatGPT | 50.20 | 51.60 |
| ChatGPT$_{(MOLWENI)}$ | 50.26 | 44.65 |
| TA-ChatGPT$_{(MOLWENI)}$ | 52.98 | 53.54 |
| RoBERTa | 72.14 | 63.61 |
| SKIER $_{w/o\ TL}$ | 73.66 | 65.87 |
| SKIER* | 75.05 | 67.39 |
| DialogDP$_{(STAC)}$ | 74.38 | 66.69 |
| DialogDP$_{(MOLWENI)}$ | 74.26 | 66.70 |
| TA-DialogDP$_{(STAC)}$ | 74.50 | 66.55 |
| TA-DialogDP$_{(MOLWENI)}$ | 74.95 | 66.84 |

Table 3: Results on the downstream ERC task. The metric used is the weighted average F1 score. The bold text reveals the best performance, and the underlined indicates the second best. Subscripts in between brackets denote the corpus used to train the parser. * means the model was fine-tuned on manually annotated relations. Here, the results of SKIER* serve as a reference point, not for direct comparison. w/o TL means without manually annotated relations.

## 4.7 Case Study

As shown in Fig 4, our model surpasses ChatGPT in L&R. In detail, A(0)->B(2), A(0)->D(4), D(4)->D(6) do not appear in ground truth but are indicated by ChatGPT. Moreover, ChatGPT fails on all QAP (label 1) pairs. The attention scores of DialogDP exhibit a notable distinction between correct links (dark) and incorrect links (light). Compared with DeepSequential, our graph-based DialogDP avoids sequential error propagation through a parallel attention mechanism.

## 4.8 Effect of Dialogue Length

We examined the effect of dialogue length on the parsing performance. As shown in Fig. 5 (a), the trend on STAC is a gradual decline in performance as the dialogue length increases. The reason is that long-range dialogues encompass a greater number of links and relation types, resulting in a more complicated parsing tree. Consequently, the performance of the parser is adversely affected, resulting in lower F1 scores. Additionally, the performance of L&R declines more rapidly compared to that of Link prediction. This suggests that the existence of long-term dependencies also presents significant challenges for the relation classifier. In Fig. 5 (b), we did not observe a comparable trend, as the length of Molweni dialogues demonstrates a concentrated distribution within a limited range of [7, 14]. This may benefit the parser.

## 4.9 Effect of Dialogue Turn

We further investigated the performance of baselines and our DialogDP at different dialogue turns on the Molweni dataset. In Fig. 6, we observed

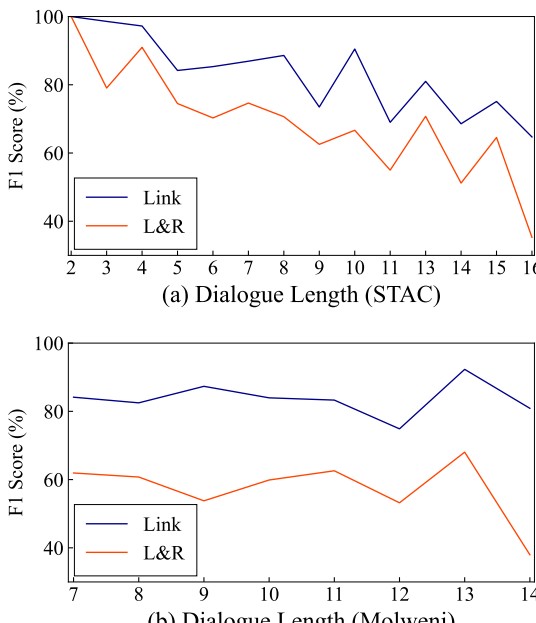

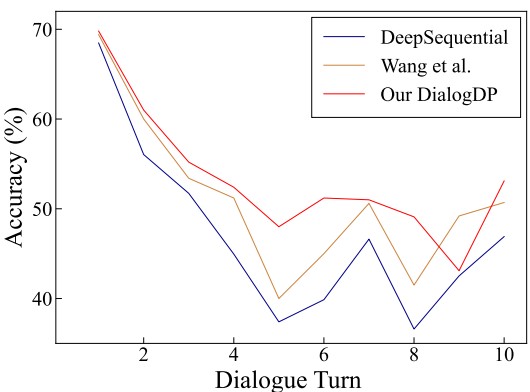

Figure 6: Comparison of prediction accuracy between typical methods at different dialogue turns.

Figure 5: The predicted F1 scores of DialogDP on the *STAC* and *Molweni* testsets.

a downward trend in all models. As the dialogue turn increases, the search space of bottom-up-based methods expands, leading to a decline in the accuracy of L&R prediction. *DeepSequential* displays the consistently lowest accuracy across all dialogue turns, largely attributed to the issue of error propagation. Meanwhile, the other two models eliminated this issue as they avoided using historical predictions for processing the current one. Different from the model of Wang et al. (2021) in Fig. 6, DialogDP shows a gentle-slope downward trend. Our method exceeds the model of Wang et al. (2021) in all the dialogue turns except the 9th turn. Additionally, DialogDP has two local minimum points at the 5th turn (48.0%) and 9th turn (43.1%), both of which performs better than those at the 5th (40.0%) and 8th (41.5%) turns. Given that the model proposed by Wang et al. (2021) employed graph-based techniques (Cai and Lam, 2020) to mitigate error propagation, our model demonstrates superior performance in addressing this issue since the bidirectional self-supervision improves the stability of prediction results by jointly utilizing the top-down and bottom-up dependency parsing graphs.

## 5 Conclusion

In this paper, we proposed a bidiretional self-supervised dialogue discourse parser DialogDP and

a task-aware paradigm combining DDP and downstream task ERC. First, we designed two graph-based parsers leveraging both bottom-up and top-down parsing strategies and eliminating sequential error propagation. Then a bidirectional self-supervision mechanism is designed to reduce the learning bias by exploiting the structural symmetry of two strategies, and thus avoid the reliance on external knowledge. Furthermore, a soft-window triangular mask, tailored with statistical information, is utilized to effectively handle long-term dependencies. Second, we presented a task-aware paradigm bridging the gap between the foundational DDP with downstream tasks through fine-tuning DialogDP with task-aware dependency relation annotations. Empirical studies on DDP and downstream ERC show the superiority and adaptability of our DialogDP and task-aware paradigm.

## Limitations

Due to the lack of theoretical support, it is challenging to design task-aware dependency relations. The design of task-specific dependency relations is expected to exert a substantial influence on the performance of downstream tasks that are integrated with DDP. Hence, it is encouraged to undertake theoretical analysis in order to devise task-specific relations that are better suited for the intended downstream tasks. This paper primarily concentrates on the ERC task to serve as an illustrative example for validating the effectiveness of the task-aware paradigm. However, additional comprehensive evaluations would be beneficial for thoroughly assessing the proposed task-aware paradigm.

## Acknowledgements

This research is supported by the Agency for Science, Technology and Research (A*STAR) under its AME Programmatic Funding Scheme (Project #A18A2b0046).

## Ethics Statement

The research conducted strictly adheres to ethical guidelines of EMNLP, ensuring privacy, fairness, transparency, responsible use, and continuous improvement.

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

# A   Statistics of datasets

In this section, we display the statistical information of *STAC* and *Molweni* datasets. The x-axis represents the distance between a pair of dependent utterances, and the y-axis represents the proportions of different distances. It is evident that the distance between dependent utterances follows a long-tailed distribution. In STAC training set, more than 90% of the dependencies are within 10 dialogue turns; More than 99% of the dependencies are within 20 dialogue turns; No dependency exists beyond 26 dialogue turns. Hence, we set hyper-parameters $B_0 = 10$, $B_1 = 20$, $B_2 = 89$ for task on the STAC dataset. Similarly, we set hyper-parameters $B_0 = 8$, $B_1 = 10$, $B_2 = 14$ for task on the Molweni dataset.

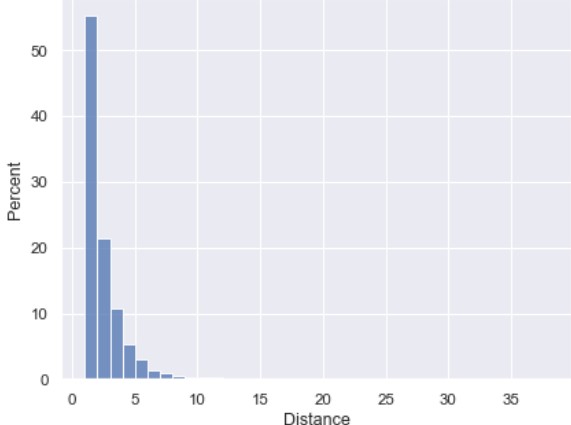

Figure 7: The distance distribution of STAC dataset

The pre-defined 16 dependency relations as shown as below:

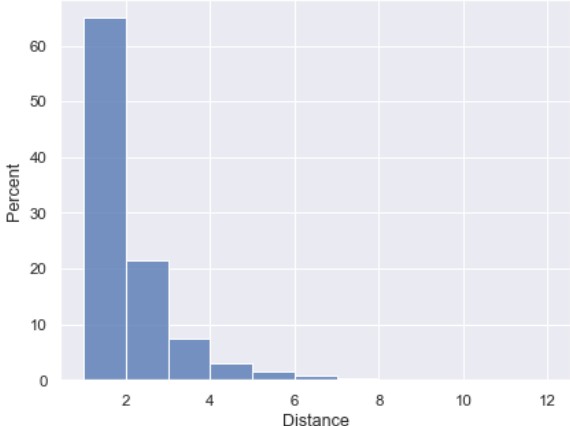

Figure 8: The distance distribution of Molweni dataset

| Dependency Relations | | |
|---|---|---|
| Comment | Q-Elab | Conditional |
| Acknowledgement | Elaboration | Background |
| Contrast | Alternation | Narration |
| Correction | Parallel | Continuation |
| QAP | Explanation | Result |
| Clarification_Q | | |

Table 4: The Predefined Dependency Relations

## B  The application of ChatGPT to DDP task

This section details the usage of ChatGPT on the DDP task. To prompt ChatGPT to parse a dialogue based on specific requirements, we begin by providing an example dialogue that includes pre-defined dependency relations and a parsed dependency tree. Subsequently, we input a dialogue where we ask ChatGPT to parse it accordingly. Below is the prompting example:

> Given the dialogue history, please predict the discourse parsing based their semantic relevance and logic flow as follows: speaker: A, text: i have enough to build a settlement too now... ?, turn: 0
> speaker: A, text: and it won't let me , turn: 1
> speaker: B, text: because you have no good roads!, turn: 2
> speaker: C, text: youre like me, turn: 3
> speaker: C, text: you need to build another road, turn: 4
> speaker: C, text: it needs to be at least 2 roads away from another of your settlements, turn: 5
> speaker: A, text: okeydoke, turn: 6
> speaker: A, text: thanks guys, turn: 7

There are 16 pre-defined dependency relations. They are ['Continuation', 'Explanation', 'Comment', 'Clarification_question', 'Question-answer_pair', 'Correction', 'Contrast', 'Acknowledgement', 'Background', 'Result', 'Elaboration', 'Conditional', 'Narration', 'Q-Elab', 'Parallel', 'Alternation'] predicted dependencies are in the form of: (0,1, relation: Contrast), (0,2, relation: Explanation), (0,3, relation: Alternation), (3,4, relation:Elaboration), (4,5, relation: Continuation), (3,6, relation: Acknowledgement), (2,7, relation: Acknowledgement), (3,7, relation: Acknowledgement).

## C  The application of ChatGPT to ERC task with dependencies from task-aware DialogDP

Input of System content: "You are an expert in sentiment analysis. The given head utterances may influence the emotion of the current utterance. Please identify the emotion label for the current utterance with one of the pre-defined emotion labels. The emotion labels are [neutral, surprise, fear, sadness, joy, disgust, anger]."

Task-aware dependency-based prompt: prompt = "Dialogue History: $U$, Relation: $r$, Head: $u_j$, Current utterance: $u_i$, Emotion label for current utterance is:".

