# OpenReview forum: "Task-Aware Self-Supervised Framework for Dialogue Discourse Parsing"
_EMNLP/2023/Conference — EMNLP 2023 Findings_

### Official Review · Reviewer_yNEK · 2023-07-29

**Soundness:** 3

**Excitement:**

2: Mediocre: This paper makes marginal contributions (vs non-contemporaneous work), so I would rather not see it in the conference.

**Missing References:**

A Distance-Aware Multi-Task Framework for Conversational Discourse Parsing, COLING, 2022.

Structured Dialogue Discourse Parsing, SIGDIAL, 2022.

**Paper Topic And Main Contributions:**

This paper proposed a task-aware self-supervised framework for dialogue discourse parsing. It designed a graph-based model to alleviate error propagation. Besides, a bottom-up and top-down parsing strategy was introduced to alleviate the problem of learning bias. Empirical studies on dialogue discourse parsing datasets and a downstream task demonstrate the effectiveness and flexibility of the proposed framework.

**Questions For The Authors:**

A. In ChatGPT baseline, how to select the demonstrations?  How to deal with the occasional failure of ChatGPT to follow instructions?

**Reasons To Accept:**

A task-aware self-supervised framework is proposed for dialogue discourse parsing.

Experimental results show the effectiveness of the proposed framework.

**Reasons To Reject:**

This paper has the following weaknesses:

A. The example in Figure 1 lacks the data source. It does not come from either Molweni or STAC used in this paper.

B. Lack of a clear motivation. This paper argued that there are three problems, including error propagation, learning bias, and incompatibility of predefined relations, that need to be addressed. However, it doesn't give a clear explanation of why these problems need to be solved.  For example, why is learning bias an issue worth considering, given that Molweni and STAC have few multi-head dependencies? What impact does the incompatibility of predefined relations have on downstream tasks?

C. Unclear experimental settings. In Section 3.3, how to solve the BERT token limitation, since the whole dialogue is concatenated? In dataset processing, if you followed Shi et al. to pre-process a dataset that doesn't consider multi-head dependencies, what is the effect of your top-down and bottom-up strategy that considers multi-head dependencies?

D. Unclear description in Section 3.8. What is the motivation for transition? Why does the existing relation taxonomy may not fit the ERC tasks? Why could this transition better facilitate ERC tasks? What principles is the transition based on?

E. Unfair comparison. This paper adopted the BERT-large, which is not fair to directly compare to the works adopting the base version. Therefore, it is not clear whether the performance improvement comes from the proposed method or from using a larger pre-trained model.

F. An in-depth analysis of each component in the proposed method is required. Since top-down and bottom-up are used to alleviate the problem of learning bias, does this strategy better identify multi-head dependencies? Since SWM is devised to alleviate the problem of long-range dependencies, does it help identify longer-distance dependencies? Besides, what problems in ERC are solved by TA-DialogDP? The overall performance is not convincing and an in-depth analysis is required.

**Reproducibility:**

2: Would be hard pressed to reproduce the results. The contribution depends on data that are simply not available outside the author's institution or consortium; not enough details are provided.

**Reviewer Confidence:**

5: Positive that my evaluation is correct. I read the paper very carefully and I am very familiar with related work.

---

> ### Author Rebuttal · Authors · 2023-08-28
>
> **We are grateful for your valuable comments, which will definitely help improve our work. We tried our best to avoid any misunderstandings. It might take you some time to read the response. We sincerely hope you could re-consider our paper after your reading.**
>
> **A. The example in Figure 1 lacks the data source. It does not come from either Molweni or STAC used in this paper.**
>
> Re: Thanks for your comment. We should have added the data source of the example. The example in Figure 1 is used to illustrate the dialogue dependency graph. Hence, we select a conversation snippet (the id of this conversation is tr\_c2969) from dailydialog dataset (a widely used English dialogue dataset) with the parsing information by Shi and Huang (2019) [1]. We manually checked the parsed dependency graph. We will add the data source of this example in the paper.
>
> [1] Shi Z, Huang M. A deep sequential model for discourse parsing on multi-party dialogues. In Proceedings of the AAAI Conference on Artificial Intelligence 2019 Jul 17 (Vol. 33, No. 01, pp. 7007-7014).
>
> **B. Lack of a clear motivation. This paper argued that there are three problems, including error propagation, learning bias, and incompatibility of predefined relations, that need to be addressed. However, it doesn't give a clear explanation of why these problems need to be solved. For example, why is learning bias an issue worth considering, given that Molweni and STAC have few multi-head dependencies? What impact does the incompatibility of predefined relations have on downstream tasks?**
>
> Re: Thanks for your valuable comment. We will explain this question from the following three aspects and will revise the paper according to your suggestions.
>
> _Response to question B (1) (error propagation)_:
>
> We mentioned in the second paragraph of the Introduction that error propagation is caused by `sequentially scanning` the utterances in dialogues and subsequently predicting dependency links and corresponding relation types, which is believed to be prone to a severe error propagation issue [1]. Taking paper of (Shi and Huang) [2] as an example, if the wrong head is selected for a dependent utterance then the wrong head would have a negative effect on the representation of the dependent utterance and the sequential scan mechanism makes this error propagate to the following utterances. This error propagation problem has been defined in paper [1] and paper [1] made efforts to alleviate this issue. In our paper, we avoid the error propagation problem by the graph mechanism.
>
> _Response to question B (2) (learning bias)_:
>
> The existing models that select only one head from multiple heads for training introduced false information. Our model is designed to fully utilize multi-head dependencies for training, instead of distorting the labels. This can be applied to any discoursing parsing task of which the newly constructed dataset contains multiple heads, not only STAC and Molweni. Additionally, according to paper [4], these long-tailed data are worth studying; Overlooking the long-tailed data may result in a biased distribution estimation during training [5].
>
> In this paper, we mainly consider two types of learning biases, i.e., `measurement bias` and `algorithmic bias` [6]. Shi and Huang[2] selected only one head from multiple heads for each dependent during the training process. This inevitably caused measurement bias since some true head labels were set to `false` labels, introducing `errors` to the training data. Our method takes multiple heads into account during the training process. Our model fundamentally avoids such `measurement bias` through the aforementioned training setting. Additionally, as we mentioned in the paper, the bidirectional self-supervision mechanism serves as regularization which helps to minimize the `algorithmic bias`.
>
> _Response to question B (3) (incompatibility)_:
>
> According to paper [3], many existing relations(e.g., narration, background) `do not/ or rarely` exist in the ERC datasets. Meanwhile, we found that a relation type may correspond to multiple sentiment shifts. For example, question\_answer\_pair may correspond to negative\_to\_positive, positive\_to\_negative, neutral\_to\_negative, etc., which may not provide sufficient information for the ERC task. In this case, the existing relations are not compatible with the ERC task. Moreover, we want to capture the emotion shift in the dependency graph. In summary, it is essential to design a mechanism that is compatible with downstream tasks.
>
> [1] Ante Wang, Linfeng Song, Hui Jiang, Shaopeng Lai, Junfeng Yao, Min Zhang, and Jinsong Su. 2021. A structure self-aware model for discourse parsing on multi-party dialogues. In Proceedings of the Thirtieth International Joint Conference on Artificial Intelligence, IJCAI-21, pages 3943–3949. International Joint Conferences on Artificial Intelligence Organization.
>
> [2] Shi Z, Huang M. A deep sequential model for discourse parsing on multi-party dialogues. In Proceedings of the AAAI Conference on Artificial Intelligence 2019 Jul 17 (Vol. 33, No. 01, pp. 7007-7014).
>
> [3] Li, W., Zhu, L., Mao, R., \& Cambria, E. (2023). Skier: A symbolic knowledge integrated model for conversational emotion recognition. In Proceedings of the AAAI Conference on Artificial Intelligence.
>
> [4] S.Bengio. The battle against the long tail.Talk on Workshop on Big Data and Statistical Machine Learning. 1, 2.
>
> [5] Wang C, Gao S, Wang P, Gao C, Pei W, Pan L, Xu Z. Label-aware distribution calibration for long-tailed classification. IEEE Transactions on Neural Networks and Learning Systems. 2022 Oct 25(99):1-3.
>
> [6] Mehrabi, N., Morstatter, F., Saxena, N., Lerman, K., \& Galstyan, A. (2021). A survey on bias and fairness in machine learning. ACM computing surveys (CSUR), 54(6), 1-35.
>
> **C. Unclear experimental settings. In Section 3.3, how to solve the BERT token limitation, since the whole dialogue is concatenated? In dataset processing, if you followed Shi et al. to pre-process a dataset that doesn't consider multi-head dependencies, what is the effect of your top-down and bottom-up strategy that considers multi-head dependencies?**
>
> Re: Thanks for the question. There are two or three conversations that exceed the token limitation of BERT. We removed punctuation and stop words to make the conversations fit for the token limitation. We mentioned that the code would be available for public access and more processing details would be released in the code repository. Also, we will include more experimental settings in the appendix.
>
> The pre-processed dataset by Shi et al. contains multi-head dependencies. However, in their DeepSequential model, they just select one head for loss calculation. We consider multi-heads during training to `fully exploit` this useful dependent-head label information which is usually ignored by the other methods. However, data with few multi-head labels (6\% multi-head labels in STAC) are not sufficient/enough to train a multi-label classifier. Besides, considering that baseline methods select one head for each dependent, we select one head in the decoding stage.
>
> **D. Unclear description in Section 3.8. What is the motivation for transition? Why does the existing relation taxonomy may not fit the ERC tasks? Why could this transition better facilitate ERC tasks? What principles is the transition based on?**
>
> Re: Thanks for your valuable comments. We explain this question from the following two aspects and will add more theoretical justifications according to your comments.
>
> _Responses to question D (1)(3)(4)_:
>
> The state transition mapping aims to represent the `emotion shift` between two utterances which are connected by a dependency link. According to the following two research papers [1][2], emotion shift detection can `enhance` the performance of ERC and is defined as a task in the sentiment analysis domain. Therefore, we believe that state transition can facilitate the ERC task. We will include the theoretical support in subsection 3.8 to justify our motivation and principle for state transition (emotion shift). Besides, research [3] in the psychology domain also implements similar methods to represent the state transition (emotion shift).
>
> _Responses to question D (2)_:
>
> The existing relation taxonomy can be applied to multiple downstream NLP tasks [4] including question-answering, dialogue summarization, and machine reading comprehension, which means the taxonomy is `not specifically tailored` for the ERC task. According to [5], many relations(e.g., narration, background) `do not/ or rarely exist` in the ERC datasets. Meanwhile, we found that a relation type may correspond to `multiple` sentiment shifts. For example, question\_answer\_pair may correspond to negative\_to\_positive, positive\_to\_negative, neutral\_to\_negative, etc., which may not provide sufficient information for the ERC task. Moreover, we want to capture the emotion shift in the dependency graph. Hence, we design a task-aware mechanism with the proposed relation taxonomy. We will add more explanations in the first paragraph of subsection 3.8 for this question.
>
> [1] Agarwal, H., Bansal, K., Joshi, A., \& Modi, A. (2022). Shapes of Emotions: Multimodal Emotion Recognition in Conversations via Emotion Shifts. ACL 2022, 44.
>
> [2] Gao, Q., Cao, B., Guan, X., Gu, T., Bao, X., Wu, J., Liu B., \& Cao, J. (2022). Emotion recognition in conversations with emotion shift detection based on multi-task learning. Knowledge-Based Systems, 248, 108861.
>
> [3] Winkler, J. R., Appel, M., Schmidt, M. L. C., \& Richter, T. (2023). The experience of emotional shifts in narrative persuasion. Media Psychology, 26(2), 141-171.
>
> [4] Li, J., Liu, M., Qin, B., \& Liu, T. (2022). A survey of discourse parsing. Frontiers of Computer Science, 16(5), 165329.
>
> [5] Li, W., Zhu, L., Mao, R., \& Cambria, E. (2023). Skier: A symbolic knowledge integrated model for conversational emotion recognition. In Proceedings of the AAAI Conference on Artificial Intelligence.
>
> **E. Unfair comparison. This paper adopted the BERT-large, which is not fair to directly compare to the works adopting the base version. Therefore, it is not clear whether the performance improvement comes from the proposed method or from using a larger pre-trained model.**
>
> Re: Thanks for the comment. We carefully checked the experimental setting of the baselines. We found that He et al.(2021) utilized `bert-large` and employed `multitask learning` to enhance the performance of dialogue parsing; Yang et al.(2021) attributed the good performance to the `joint training` of conversational discourse parsing and dropped pronoun recovery tasks; Yu et al.(2022) leveraged `external corpus` (the authors mentioned ``We pre-train BERT on a large-scale unlabeled dialogue corpus in the second stage. It is collected from the Ubuntu dialogue corpus (Lowe et al., 2015), containing 930,000 unlabeled dialogues.''). Our model did not introduce external knowledge/ corpus nor leverage a different auxiliary task. In this case, we think it is fair to compare our model with these baselines.
>
> **F. An in-depth analysis of each component in the proposed method is required. Since top-down and bottom-up are used to alleviate the problem of learning bias, does this strategy better identify multi-head dependencies? Since SWM is devised to alleviate the problem of long-range dependencies, does it help identify longer-distance dependencies? Besides, what problems in ERC are solved by TA-DialogDP? The overall performance is not convincing and an in-depth analysis is required.**
>
> Re: Thanks for your valuable comments. We explain this question from the following three aspects and will add more analysis of each component in the experiment section.
>
> _Response to question F (1)_:
>
> In this paper, we mainly consider two types of learning biases, i.e., `measurement bias` and `algorithmic bias` [1]. Shi and Huang[2] selected only one head from multiple heads for each dependent during the training process. This inevitably caused measurement bias since some true head labels were set to false labels, introducing errors to the training data. Our method takes multiple heads into account during the training process. Our model fundamentally avoids such `measurement bias` through the aforementioned training setting, which does not need extra experiment analysis. Additionally, as we mentioned in the paper, the bidirectional self-supervision mechanism serves as regularization which helps to minimize the `algorithmic bias`. The ablation study showed the effectiveness of the mechanism through the results in the 1st, 2nd, 3rd, 4th, 7th and 8th lines of Table 2.
>
> The bottom-up and top-down strategies are used for self-supervision, thus `reducing the algorithmic bias`. They are not used for identifying multi-head dependencies. According to Shi and Huang (2019)[1], only around 6\% of the utterances have multiple heads. Although multi-head dependencies are considered during the training process, data with 6\% percent multi-head labels are not sufficient/enough to train a multi-label classifier.
>
> _Response to question F (2)_:
>
> The SWM helps identify long-distance dependencies. This can be verified by the comparison between our model and two baselines in papers [2] and [3]. In paper [2], the authors set a fixed window and ignored the dependencies outside the window, which means their model fails to capture long-distance dependencies. Comparing Figure 6 in paper [3] and Figure 5 in our paper, it can be observed that the performance of the model in paper[3] on STAC drops below 50\% as the dialogue lengths increase to 13; while our model maintains the performance above 50\% until the dialogue lengths increase to 16. Similarly, the better performance in identifying long-distance dependencies of our model can be observed on the Molweni dataset.
>
> _Response to question F (3)_:
>
> TA-DialogDP captures the state transition which can be used by the models(e.g., SKIER) employing the parsed graph on the downstream ERC task.
>
> On average, our proposed methods outperform SKEIR without TL by 0.83 and 0.86 in terms of 7-cls and 3-cls, respectively. SKEIR without TL utilized a parsed graph with existing relation taxonomy while our method (TA-DialogDP) employed a parsed graph with task-aware relation labels. SKIER\* denotes that the model was fine-tuned on `human-annotated` links and relations. Our TA-DialogDP achieved results comparable with SKIER\* although it was not finetuned with human-annotated links and relations on downstream dataset.
>
> We will add an in-depth analysis to illustrate how our model alleviates algorithmic bias, why it is skilled at identifying long-distance dependencies, and how the ERC model was enhanced by the task-aware mechanism.
>
> [1] Mehrabi, N., Morstatter, F., Saxena, N., Lerman, K., \& Galstyan, A. (2021). A survey on bias and fairness in machine learning. ACM computing surveys (CSUR), 54(6), 1-35.
>
> [2] Shi Z, Huang M. A deep sequential model for discourse parsing on multi-party dialogues. In Proceedings of the AAAI Conference on Artificial Intelligence 2019 Jul 17 (Vol. 33, No. 01, pp. 7007-7014).
>
> [3] Yu, N., Fu, G., \& Zhang, M. (2022, October). Speaker-Aware Discourse Parsing on Multi-Party Dialogues. In Proceedings of the 29th International Conference on Computational Linguistics (pp. 5372-5382).
>
> **A. In ChatGPT baseline, how to select the demonstrations? How to deal with the occasional failure of ChatGPT to follow instructions?**
>
> Re: Thanks for your question.
>
> _Response to question A (1)_:
>
> We are sorry that we are not sure about what the demonstrations refer to. We will release the code of the whole project including the experiments of ChatGPT baseline.
>
> _Response to question A (2)_:
>
> We designed a mechanism based on regular expressions to examine the retrieved results. The algorithm will repeatedly retrieve the response for the same input until the result passes the examination mechanism.
>
> **Missing References:
> A Distance-Aware Multi-Task Framework for Conversational Discourse Parsing, COLING, 2022. Structured Dialogue Discourse Parsing, SIGDIAL, 2022.**
>
> Re: Thanks for your suggestion. We will cite and discuss these papers in the related work section.

---

### Official Review · Reviewer_jNsR · 2023-07-31

**Soundness:** 3

**Excitement:**

3: Ambivalent: It has merits (e.g., it reports state-of-the-art results, the idea is nice), but there are key weaknesses (e.g., it describes incremental work), and it can significantly benefit from another round of revision. However, I won't object to accepting it if my co-reviewers champion it.

**Paper Topic And Main Contributions:**

This paper proposes a task-aware self-supervised framework for dialogue discourse parsing, which is a fundamental natural language processing task. The paper addresses the issue of existing parsing approaches being constrained by predefined relation types, which can impede the adaptability of the parser for downstream tasks. The main contributions of the paper are:

1. Introducing a task-aware paradigm to improve the versatility of the dialogue discourse parser for downstream tasks.
2. Designing a graph-based discourse parsing model called DialogDP to alleviate error propagation and learning bias.
3. Developing an innovative self-supervised mechanism that leverages both bottom-up and top-down parsing strategies to allow the parsing graphs to mutually regularize and enhance each other.
4. Demonstrating the effectiveness and flexibility of the proposed framework through empirical studies on dialogue discourse parsing datasets and a downstream task.

In summary, this paper proposes a novel approach to dialogue discourse parsing that improves the adaptability of the parser for downstream tasks and addresses issues with existing parsing approaches.

**Reasons To Accept:**

The strengths of this paper are:

1. Introducing a task-aware paradigm to improve the versatility of the dialogue discourse parser for downstream tasks.
2. Designing a graph-based discourse parsing model called DialogDP to alleviate error propagation and learning bias.
3. Demonstrating the effectiveness and flexibility of the proposed framework through empirical studies on dialogue discourse parsing datasets and a downstream task.

If this paper were to be presented at the conference or accepted into Findings, the main benefits to the NLP community would be:

1. Advancing the state-of-the-art in dialogue discourse parsing and improving the adaptability of the parser for downstream tasks.
2. Providing a novel approach to address issues with existing parsing approaches and demonstrating the effectiveness and flexibility of the proposed framework through empirical studies.

**Reasons To Reject:**

1. The paper lacks sufficient analysis of the experimental results, especially on the Molweni dataset.
2. There is some overlap between the relevant work section and the introduction section.

**Reproducibility:**

4: Could mostly reproduce the results, but there may be some variation because of sample variance or minor variations in their interpretation of the protocol or method.

**Reviewer Confidence:**

4: Quite sure. I tried to check the important points carefully. It's unlikely, though conceivable, that I missed something that should affect my ratings.

---

> ### Author Rebuttal · Authors · 2023-08-28
>
> **We are grateful for your valuable comments, which will definitely help improve our work.**
>
> **1. The paper lacks sufficient analysis of the experimental results, especially on the Molweni dataset.**
>
> Re: Thanks for your valuable comment. Both the main results (Table 1) and the Ablation Study (Table 2) actually directly demonstrate the model's performance on STAC and Molweni. Section 4.6, TA-DialogDP on the ERC task, indirectly showcases the contributions of TA-DialogDP trained on STAC and Molweni to downstream tasks. Since the results on both datasets are similar, we mainly analyze the `overall trend` to keep the explanation concise. In Section 4.8, the results from the two datasets are presented separately due to their notable differences. Nevertheless, we will include an in-depth analysis of the results from the Molweni dataset in subsections 4.4, 4,5 and 4.6.
>
> **2. There is some overlap between the relevant work section and the introduction section.''**
>
> Re: Thanks for your valuable comment. We will revise the Introduction and related work to avoid overlap. Specifically, we will remove some references that already appeared in the introduction and add reviews on other relevant research in the related work section.

---

### Official Review · Reviewer_okwz · 2023-08-10

**Soundness:** 4

**Excitement:**

3: Ambivalent: It has merits (e.g., it reports state-of-the-art results, the idea is nice), but there are key weaknesses (e.g., it describes incremental work), and it can significantly benefit from another round of revision. However, I won't object to accepting it if my co-reviewers champion it.

**Missing References:**

N/A

**Paper Topic And Main Contributions:**

In this paper, the authors presents a new framework for task-aware dialog discourse parsing. The framework involves a biaffine attention mechanism to produce probabilities for a dependency to be present between two utterances as well as the types of dependencies, then applies a soft-window triangular mask to mask the probabilities before predicting the final dependencies and types via softmax over the probabilities. The main novelties of the framework includes a self-supervision objective between separate top-down and bottom-up biaffine attention outputs, and using a soft-window mask based on relative utterance distance instead of a hard window. The authors conducted experiments on two dialog discourse parsing datasets: STAC and Molweni, and outperforming all baselines when evaluated on Link and Relation correctness. The authors also performed ablation studies and demonstrated the ability to extend the trained framework to downstream tasks such as ERC.

**Questions For The Authors:**

(A) Does the self supervision part happen before the triangular mask or after the triangular mask? In text it is presented before the triangular mask section, but in figure 2 the self-supervision arrow is shown after the soft-window triangular mask.

(B) What are Ua and Ul in equations 1 and 2? There isn't any text description of their definitions. Are they relevant to the U defined in line 223? In general, the uses of all these U in the 2 equations are quite confusing, as some of them are upper-case and some are lower case, some are italicized and some are not, and the casing/italicizing do not match with those corresponding parts in Figure 2.

(C) What is the definition of dij in equation on line 307? Is it simply | i - j |?

(D) In section 3.7, why do we only allow choosing one head if sometimes there are multiple ground-truth heads? Since we are already training it as a multi-label classification problem, why can't we choose the heads via multi-label classification?

(E) In section 4.6, what do you mean by "using SKIER as backbone"? Do you mean using it as a baseline? Also, in Table 3, is SKIER* the only method that uses manually annotated relations for finetuning?

**Reasons To Accept:**

The paper presents a new framework for dialog discourse parsing, with several interesting novel ideas including a self-supervision objective between top-down and bottom-up approaches and a soft window mask. These ideas could inspire new approaches not only within DDP but also DAG-parsing tasks in general.

The authors demonstrated the strength of their framework in the experiments, as the framework outperformed all baselines in accuracy of Link and Relations on two datasets. The new framework also outperformed baselines in downstream ERC task, thus demonstrating the benefit of task-aware setups.

The authors conducted comprehensive ablation studies to justify the various design choices for the framework.

**Reasons To Reject:**

Reproduction of the method could be difficult as some key methodology details and hyperparameters are missing. For example, the lambda_a and lambda_l of the objective (equation 12) are not provided in the main text or in appendix. I included more questions about some methodology details in the Questions section.

While the new framework did achieve great performance in both experiments, the improvements are quite marginal (especially in STAC and Molweni experiments) and thus made the results less exciting.

While the ablation studies justified many design choices (such as using the soft window), some choices still seems quite arbitrary, such as selection of specific values of a0,a1,a2,B0,B1,B2 for the soft window.

**Reproducibility:**

3: Could reproduce the results with some difficulty. The settings of parameters are underspecified or subjectively determined; the training/evaluation data are not widely available.

**Reviewer Confidence:**

2: Willing to defend my evaluation, but it is fairly likely that I missed some details, didn't understand some central points, or can't be sure about the novelty of the work.

**Typos Grammar Style And Presentation Improvements:**

Line 134 and 393  "DPP" should be "DDP"

Equation 1,2 and Figure 2: The casing and italicizing of "Ua" and "Ul" are all different and not consistent.

Some parts of Figure 3 are quite confusing. For example, why is there an arrow from dependency to D/S task with an "X" over it? What does the backwards arrow from D/S task to Task-aware DialogDP stand for? Perhaps adding some explanations to the caption will help.

---

> ### Author Rebuttal · Authors · 2023-08-28
>
> **We are grateful for your valuable comments, which will definitely help improve our work.**
>
> **Reproduction of the method could be difficult as some key methodology details and hyperparameters are missing. For example, the lambda\_a and lambda\_l of the objective (equation 12) are not provided in the main text or in appendix. I included more questions about some methodology details in the Questions section.**
>
> Re: Thanks for your comment. As we mentioned in the first footnote, the code will be available for public access once accepted. Besides, we will include more details in the appendix to help readers reproduce/ understand our proposed framework. lambda\_a and lambda\_l are trade-off parameters and are set to 0.05 in this paper.
>
> **While the new framework did achieve great performance in both experiments, the improvements are quite marginal (especially in STAC and Molweni experiments) and thus made the results less exciting.**
>
> Re: Thanks for your comment. Since dialogue discourse parsing is a very challenging task, it is hard to gain significant performance improvements. For example, the performance improvements of the baseline methods are quite limited (e.g., Yu et al. 2022, Yang et al. 2021, Li et al. 2020). Even though, we still observe a relatively significant improvement on the STAC dataset (`1.1\% in L\&R`). In general, our method demonstrates a slightly better performance than the strongest baseline Yu et al. 2022. On the other hand, the main contributions we have claimed are the self-supervision structure which improves the performance of DDP without joint training of the auxiliary task and external corpus, and the task-aware mechanism which benefits downstream tasks. Incorporating the auxiliary task or an external corpus would lead to additional enhancements in the performance of our proposed approach. As for the experiments on the downstream ERC task, it is worth noting that the SKIER with a \* mark denoting that the model was fine-tuned on `human-annotated` relations. Consequently, a direct comparison between the outcomes of our approaches and SKIER* would be inequitable. On average, our proposed methods outperform SKEIR without TL by 0.83 and 0.86 in terms of 7-cls and 3-cls, respectively.
>
> **While the ablation studies justified many design choices (such as using the soft window), some choices still seems quite arbitrary, such as selection of specific values of a0,a1,a2, B0,B1,B2 for the soft window.**
>
> Re: Thanks for your valuable comment. As we mentioned in the Appendix A, the selection of values of a0, a1, a2, b0,b1, b2 are mainly based on the `statistical information` of the two datasets. For example, In STAC training set, more than `90\%` of the dependencies are within `10` dialogue turns; More than `99\%` of the dependencies are within `20` dialogue turns; No dependency exists beyond 26 dialogue turns. The results in Figure 5 (a) indicate that the performance drops fast when the length of dialogue exceeds 10, which coincides with our expectations. On the other hand, a parameter analysis is necessary for the selection of hyperparameters. We conducted experiments to further determine the specific values of such hyperparameters. We will give more justifications for the selection of hyperparameters in the appendix.
>
> **(A) Does the self supervision part happen before the triangular mask or after the triangular mask? In text it is presented before the triangular mask section, but in figure 2 the self-supervision arrow is shown after the soft-window triangular mask.**
>
> Re: Thanks for this question. As illustrated in Figure 2, the self-supervision part happens after the soft-window triangular mask. As for the presentation, we introduce self-supervision before the soft-window triangular mask as it is the core part of our proposed framework and the SWT mask can be seen as an essential component of the self-supervision mechanism.
>
> **(B) What are Ua and Ul in equations 1 and 2? There isn't any text description of their definitions. Are they relevant to the U defined in line 223? In general, the uses of all these U in the 2 equations are quite confusing, as some of them are upper-case and some are lower case, some are italicized and some are not, and the casing/italicizing do not match with those corresponding parts in Figure 2.**
>
> Re: Thank for your question. We are sorry that we should have added a clear description of the two notations. $u_a^{(1)}$ (dim is $1\times d$), $U_a^{(1)}$ (dim is $d \times d$) and $\textbf{U}_l^{(1)}$ (dim is $d\times|C|\times d$) in equations 1 and 2 are learnable parameters of the biaffine mechanism. They are not relevant to the U defined in line 223. We will replace $U$ and $u$ with $W$ and $w$, respectively, in equations 1 and 2 to make it clear.
>
> In this paper, we use the lowercase letter to represent a vector, use the italicized uppercase letter to represent a matrix, and use the bold uppercase letter to represent a 3-dimensional tensor.
>
> **(C) What is the definition of dij in equation on line 307? Is it simply $| i - j |$?**
>
> Re: Yes, $d_{i,j}$ is simply $| i - j |$. We will add an introduction of it.
>
> **(D) In section 3.7, why do we only allow choosing one head if sometimes there are multiple ground-truth heads? Since we are already training it as a multi-label classification problem, why can't we choose the heads via multi-label classification?**
>
> Re: Thanks for your question. According to Shi and Huang (2019), only around 6\% of the utterances have multiple heads. Almost all the previous methods select only one head during training. We consider multi-heads during training to `fully exploit` this useful dependent-head label information which is usually `ignored` by the other methods. However, data with 6\% percent multi-head labels are not sufficient/enough to train a multi-label classifier. On the other hand, as all the baselines only select one head for each dependent, we also select one head for a fair comparison.
>
> **(E) In section 4.6, what do you mean by "using SKIER as backbone"? Do you mean using it as a baseline? Also, in Table 3, is SKIER\* the only method that uses manually annotated relations for finetuning?**
>
> Re: Thanks for your question. We want to test the effect of task-aware dialogue discourse parsing on downstream task, i.e., ERC. Meanwhile, we found SKIER `explicitly leverages` the dialogue dependency graph for ERC task. Therefore, we use SKIER as a backbone to utilize the dialogue dependency graph generated by our Task-aware DialogDP for downstream ERC.
>
> Besides, we also use SKIER w/o TL and SKIER\* as the baseline models utilizing dialogue dependency graphs which contain the general dependency relations instead of task-aware dependency labels.
>
> We will add more descriptions to clarify the usage of the SKIER model. Yes, SKIER\* is the only one that uses manually annotated relations for finetuning. We present the outcome of SKIER\* not with the intention of making a comparison, but rather as a point of reference.
>
> **Line 134 and 393 "DPP" should be "DDP"**
>
> Re: We will fix it. Thank you.
>
> **Equation 1,2 and Figure 2: The casing and italicizing of "Ua" and "Ul" are all different and not consistent.**
>
> Re: We explained this in question B. $U^{(1)}$ and $u^{(2)}$ in Fig.2 corresponds to $U^{(1)}_a$ and $u^{(2)}_a$ respectively. We omitted the subscript $l$ in Fig.2 for simplicity as they appear in the Biaffine Attention Mechanism for Link prediction. We will add the subscript $l$ to make keep them consistent. Thank you!
>
> **Some parts of Figure 3 are quite confusing. For example, why is there an arrow from dependency to D/S task with an "X" over it? What does the backwards arrow from D/S task to Task-aware DialogDP stand for? Perhaps adding some explanations to the caption will help.**
>
> Re: In the previous research, the dialogue dependency graph with general relation labels is directly used for downstream task, i.e., ERC. However, in the framework of task-aware DDP, we do not use the general relation labels (corresponds to the "X" in the Figure 3); Instead, we `automatically re-annotate` relations based on the downstream task (`step (c)` in line 377). Then we `finetune` Task-aware DialogDP (`step (d)` in line 381) and predict the dependency links and relations with the finetuned task-aware DialogDP for downstream tasks.
>
> Here, the backwards arrow from D/S task to Task-aware DialogDP corresponds to the step (c) in line 377.
>
> Thanks for the suggestion. We will add the explanations to the caption and Fig.3 to facilitate the comprehension.

---

### Meta-Review · Area_Chair_b1kL · 2023-09-15

**Recommendation:** 3

**Metareview:**

This paper presents a method to do dialog discourse parsing (DDP), which is distinct from other forms of discourse parsing and benchmarks.  (The authors present this as a "fundamental NLP task", but that would mean that most other NLP tasks require it, which is not quite the case in practice.) The method includes a combination of both bottom-up and top-down parsing, producing a pair of graphs that need joint decoding.  This is aided by a singular triangular window (SWT) mechanism to provide soft constraints on the way links are established between utterances.

The reviewers rightly pointed out the results are an improvement but rather marginal.  Also, the analysis on the Molweni task/dataset could be improved.  There were also concerns about reproducibility and comparison to GNN's, some of which were addressed in rebuttal.

---

### Decision · Program_Chairs · 2023-10-07

**Decision:**

Accept-Findings

**Comment:**

This paper presents a method to do dialog discourse parsing (DDP), which is distinct from other forms of discourse parsing and benchmarks.  (The authors present this as a "fundamental NLP task", but that would mean that most other NLP tasks require it, which is not quite the case in practice.) The method includes a combination of both bottom-up and top-down parsing, producing a pair of graphs that need joint decoding.  This is aided by a singular triangular window (SWT) mechanism to provide soft constraints on the way links are established between utterances.

The reviewers rightly pointed out the results are an improvement but rather marginal.  Also, the analysis on the Molweni task/dataset could be improved.  There were also concerns about reproducibility and comparison to GNN's, some of which were addressed in rebuttal.